# A Herpesviral induction of RAE-1 NKG2D ligand expression occurs through release of HDAC mediated repression

Trever T Greene[1†], Maria Tokuyama[1†], Giselle M Knudsen[2], Michele Kunz[1], James Lin[1], Alexander L Greninger[2], Victor R DeFilippis[3], Joseph L DeRisi[2], David H Raulet[1], Laurent Coscoy[1]*

[1]Department of Molecular and Cell Biology, University of California, Berkeley, United States; [2]Department of Biochemistry and Biophysics, University of California, San Francisco, United States; [3]Vaccine and Gene Therapy Institute, Oregon Health and Science University, Beaverton, United States

*For correspondence: lcoscoy@berkeley.edu

†These authors contributed equally to this work

Competing interests: The authors declare that no competing interests exist.

**Abstract** Natural Killer (NK) cells are essential for control of viral infection and cancer. NK cells express NKG2D, an activating receptor that directly recognizes NKG2D ligands. These are expressed at low level on healthy cells, but are induced by stresses like infection and transformation. The physiological events that drive NKG2D ligand expression during infection are still poorly understood. We observed that the mouse cytomegalovirus encoded protein m18 is necessary and sufficient to drive expression of the RAE-1 family of NKG2D ligands. We demonstrate that RAE-1 is transcriptionally repressed by histone deacetylase inhibitor 3 (HDAC3) in healthy cells, and m18 relieves this repression by directly interacting with Casein Kinase II and preventing it from activating HDAC3. Accordingly, we found that HDAC inhibiting proteins from human herpesviruses induce human NKG2D ligand ULBP-1. Thus our findings indicate that virally mediated HDAC inhibition can act as a signal for the host to activate NK-cell recognition.

## Introduction

Natural Killer (NK) cells are an important part of the immune system, particularly to control viral infections and cancers (*Raulet and Guerra, 2009*; *Lodoen and Lanier, 2006*). NK cells recognize and lyse cells undergoing infection or transformation as well as secrete pro-inflammatory molecules that activate the adaptive immune system (*Smyth et al., 2005a*).

The activating receptor NKG2D is expressed on NK cells and provides one mechanism by which NK cells (as well as NK T cells, γδT cells, and CD8 and CD4 T cells) recognize their targets (*Raulet, 2003*). Ligands for NKG2D are absent or expressed at low levels on normal cells, but their expression is increased in response to infection or transformation, enabling NKG2D recognition (*Raulet and Guerra, 2009*; *Lodoen and Lanier, 2006*). NKG2D surveillance protects mice from several models of induced or spontaneous cancer (*Smyth et al., 2005b*; *Guerra et al., 2008*). In humans, a polymorphism predicted to reduce NKG2D signaling is linked to both increased cancer risk (*Roszak et al., 2012*) and susceptibility to viral infections (*Taniguchi et al., 2015*).

The ligands for NKG2D are diverse, and include two families in humans (ULBP1-6 and MICA-B). In mice, the MICA-B family is not represented and the ULBP family is subdivided into RAE-1α-ε, MULT-1, and H60a-c subfamilies (*Raulet et al., 2013*). Each of these ligands binds NKG2D with a different affinity, but expression of a single family member is sufficient to stimulate killing by NK cells (*Champsaur and Lanier, 2010*). Thus, expression of these ligands are tightly controlled and nearly absent on healthy cells. It is still not well understood how the majority of NKG2D ligands are silenced

in healthy tissue, but work by López-Soto and colleagues have shown that silencing by histone deacetylases (HDACs) is involved, at least for ULBP-1 (*López-Soto et al., 2009*). Most work on NKG2D ligand regulation has focused on their induction in cancer models. In this regard, several stimuli leading to ligand expression have been identified, these include: DNA damage (*Gasser et al., 2005*), hyper-proliferation (though the transcription factor E2F) (*Jung et al., 2012*), and PI3K/Ras signaling (*Tokuyama et al., 2011*; *Liu et al., 2012*). Understanding the exact mechanism of NKG2D ligand induction in various contexts remains an active area of research.

NKG2D activation and evasion play particularly important roles during herpesvirus infection. Over 90% of the population is infected with one or more of these double stranded DNA viruses (*Wald and Corey, 2007*; *Staras et al., 2006*), which include cytomegalovirus (CMV), herpes simplex viruses (HSV), and Epstein Barr Virus (EBV). Co-evolution between herpesviruses and their hosts has resulted in a complex network of host-virus interactions that allows the host to sustain a life-long infection; the majority of infections are asymptomatic in immunocompetent individuals (*Taniguchi et al., 2015*; *Biron et al., 1989*). Herpesvirus infection causes a transcriptional upregulation of NKG2D ligand expression (*Lodoen et al., 2003*), yet these viruses apply multiple strategies to avoid NKG2D recognition by reducing NKG2D ligand expression at the surface of infected cells (Reviewed in *Jonjić et al., 2008*). This creates a balance between NK cell activation and viral evasion, and disruption of this balance can impact the outcome of infection. Hosts deficient in NK cell responses, including those with mutations that are predicted to specifically reduce NKG2D signaling, are especially prone to pathogenic herpesvirus infection (*Taniguchi et al., 2015*; *Biron et al., 1989*). Conversely, growth of herpesviruses with a defect NKG2D ligand evasion are attenuated in vivo (*Jonjić et al., 2008*).

The mechanisms governing NKG2D ligand regulation appear to be conserved between viral and non-viral stresses. For example, Phosphoinositide 3-kinase (PI3K) signaling is required for NKG2D ligand expression in transformed cells as well as during mouse cytomegalovirus (MCMV) infection (*Tokuyama et al., 2011*). Furthermore, DNA damage signaling can mediate NKG2D ligand induction in both cancer development (*Gasser et al., 2005*) and Kaposi's sarcoma herpesvirus (KSHV) infection (*Bekerman et al., 2013*). Understanding how cells regulate NKG2D ligands in the context of herpesvirus infection should therefore provide insight into ligand regulation and NK cell activation in systems beyond herpesvirus infection.

Here, we show that in healthy cells, the expression of the RAE-1 family is repressed by the action of HDAC3 and describe a novel mechanism of ligand regulation during viral infection. We identified a single MCMV protein, m18, which is necessary and sufficient to induce expression of the RAE-1 family of mouse NKG2D ligands, and to induce NKG2D-dependent NK-cell killing in vitro. We found that m18 is a virally encoded HDAC inhibitor, and that it de-represses transcription of the *Raet1* gene. The m18 protein directly interacts with Casein Kinase II (CK2) and prevents it from activating HDAC3. Notably, we also demonstrated that other viral HDAC inhibitors encoded by human herpesviruses similarly drive NKG2D ligand expression. Thus we propose a model in which constitutive HDAC activity suppresses NKG2D ligand expression in healthy tissue. During infection herpesviruses express proteins that inhibit HDAC activity, directly or indirectly, and this drives NKG2D ligand expression.

## Results

### RAE-1 induction during MCMV infection requires the m18 ORF

Mouse fibroblasts normally express low levels of *Raet1* mRNA and protein, but RAE-1 expression is upregulated in response to cell stress, including viral infection (*Lodoen et al., 2003*). Previously, we showed that MCMV gene expression was necessary to induce RAE-1 expression during infection (*Tokuyama et al., 2011*). To identify the MCMV gene(s) responsible for this phenotype, we screened a panel of MCMV mutants (provided by Dr. Hidde Ploegh, Whitehead Institute) lacking genomic regions not essential for viral replication in vitro. We infected mouse fibroblasts with wild type (WT) or mutant MCMV and measured *Raet1* mRNA expression by RT-qPCR 24 hr post-infection (h.p.i.). In contrast to WT MCMV or other mutant viruses, the MCMV mutant virus deleted for open reading frames (ORFs) m01 through m22 (Δm1-22) failed to induce *Raet1* expression (*Figure 1A*). We assessed whether this was the result of decreased infection or replication in our samples by

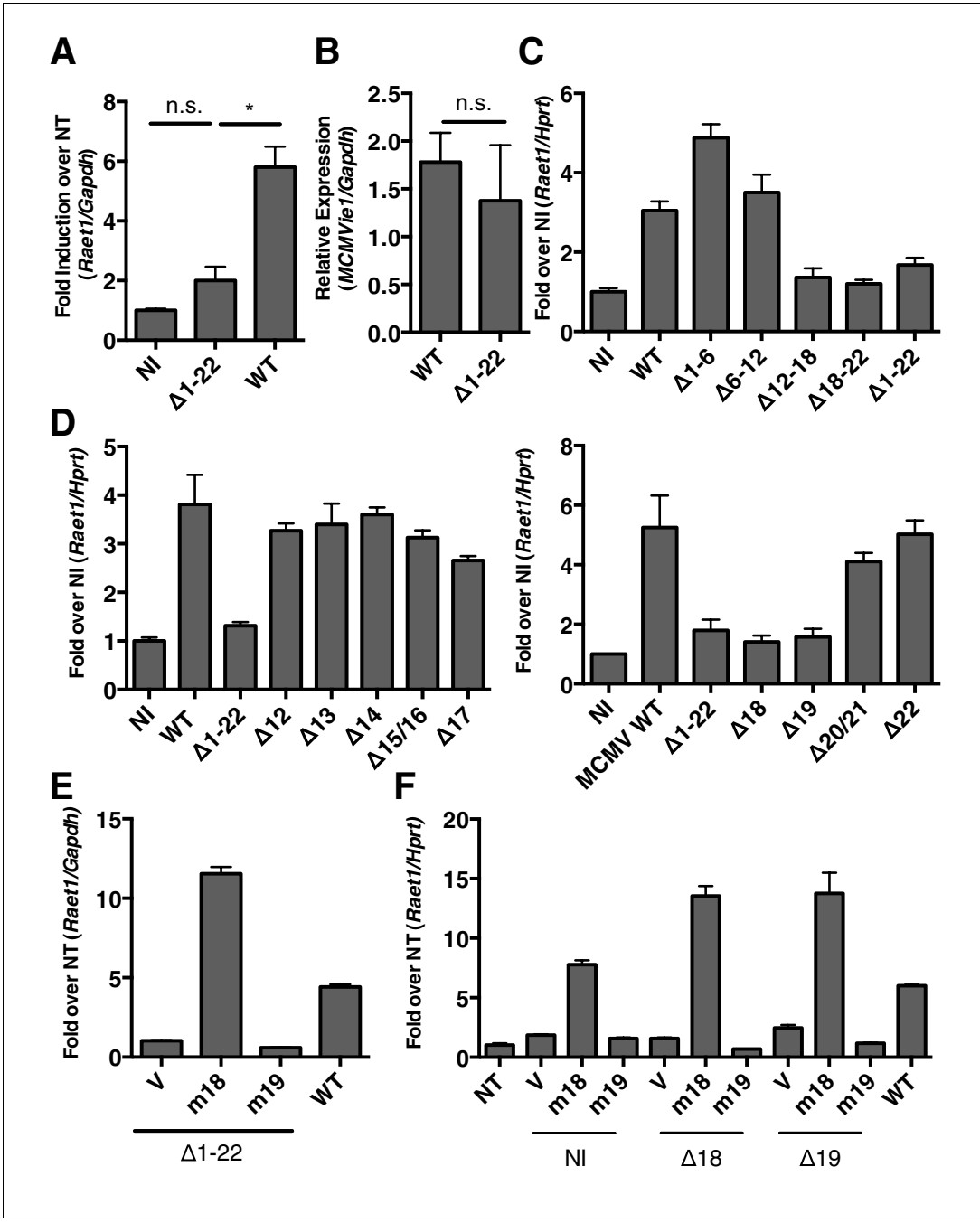

**Figure 1.** MCMV gene m18 is required for the induction of *Raet1* by MCMV. (**A**) RT-qPCR analysis of *Raet1* expression during WT and Δm1-m22 MCMV infection. Data are normalized to uninfected fibroblasts (NI). *p<0.05 n.s., not significant (1-way ANOVA with Bonferroni's multiple comparison post-test). (**B**) RT-qPCR analysis of MCMV ie1 expression during WT and Δm1-m22 infection. n.s., not significant (student's t-test). (**C, D**) RT-qPCR analysis of *Raet1* expression during infection with WT or mutant MCMV lacking the indicated ORFs. Data are normalized to uninfected fibroblasts (NI). (**E, F**) RT-qPCR analysis of *Raet1* expression in mouse fibroblasts infected with mutant MCMV lacking the indicated ORFs and subsequently transfected with m18, m19, or empty vector control. Data are normalized to uninfected fibroblasts (NI). All data were obtained by RT-qPCR and are represented as mean±SEM. All data are representative of at least three independent experiments.

measuring expression of MCMV immediate early gene 1 (ie1). Levels of MCMV ie1 were comparable to WT virus (*Figure 1B*). These results indicate that a viral gene or a combination of genes in this region is required to induce *Raet1* expression.

The region deleted in Δm1-22 MCMV contains 22 genes and several miRNAs (*Juranic Lisnic et al., 2013*). To narrow the list of candidates we generated additional MCMV mutants using bacterial artificial chromosome (BAC) recombination. We again infected mouse fibroblasts with WT and mutant viruses and measured *Raet1* expression by RT-qPCR 24 h.p.i. Viruses lacking ORFs m12 to m18 (Δ12–18) and m18 to m22 (Δ18–22) failed to induce *Raet1* expression in contrast to WT MCMV (*Figure 1C*). No known miRNAs are expressed from these deleted regions (*Juranic Lisnic et al., 2013*). To identify individual genes required for *Raet1* induction, we made individual deletions for each of the genes from m12 to m22 and again infected mouse fibroblasts with these viruses 24 hr before measuring *Raet1* expression by RT-qPCR. Only MCMV lacking ORF m18 or ORF m19 failed to induce *Raet1* expression (*Figure 1D*), indicating that one or both of these ORFs are required to induce *Raet1* expression.

To determine if expression of m18 or m19 in trans could rescue *Raet1* expression, mouse fibroblasts were first infected with WT MCMV, Δm1-m22, and Δ18 or Δ19 MCMV, and then transfected with a plasmid encoding m18, m19, or an empty control plasmid. Expression of m18, but not m19 or vector control, rescued *Raet1* expression in all cases *Figure 1E,F*) indicating that m18 is required for MCMV mediated *Raet1* induction.

## m18 expression is sufficient to induce RAE-1 expression and NK-cell killing

To assess if m18 expression is sufficient to induce *Raet1* expression we transfected mouse fibroblasts with mammalian expression plasmids encoding m18, m19, or vector control and analyzed *Raet1* mRNA by RT-qPCR (*Figure 2A*) and protein expression by flow cytometry (*Figure 2B*). Expression of m18 but not m19 was sufficient to induce *Raet1* transcript and RAE-1 protein expression. Thus, our data indicate that m18 is the both necessary and sufficient to drive RAE-1 expression.

A recent study suggests that the m19 ORF is unlikely to encode any viral mRNA or protein (*Juranic Lisnic et al., 2013*). As m19 is immediately adjacent to the transcriptional start site of m18 it is likely that Δ19 MCMV disrupts production of m18. This idea is bolstered by our data showing m18, but not m19, expression vector rescues RAE-1 expression in cells infected with Δm19 virus (*Figure 1F*). Thus we focused the rest of our study on the functional role of m18.

We next measured the ability of m18 to trigger NK-mediated killing using a chromium release assay. Fibroblasts transduced with m18 or an empty vector control were labeled with $^{51}$Cr, then incubated with IL-2 activated NK cells derived from WT or NKG2D knockout (KO) littermates at varying effector to target ratios. We observed that WT NK cells lysed m18-expressing fibroblasts in an NKG2D dependent manner (*Figure 2C*).

Previous studies have identified a CTL epitope produced from the m18 ORF (*Holtappels et al., 2002*), and peptides that correspond to this ORF have been identified in mass spectrometric analysis of MCMV virions (*Kattenhorn et al., 2004*). Otherwise little is known about m18 or its function. To characterize m18 we assessed the expression of m18 over the course of infection by harvesting RNA of MCMV infected fibroblasts at 2,4,8,12, and 24 hr post infection and amplifying the full length (3 kB) m18 mRNA by reverse transcription followed by semi quantitative PCR. In accordance with previous studies we observed this was mRNA produced as early as 2 hr post-infection (*Figure 2D*) (*Lacaze et al., 2011*). To characterize the m18 polypeptide we expressed a C-terminal hemagglutinin (HA)-tagged m18 protein in mouse fibroblasts and NIH 3T3s and analyzed cell lysates by western blot (*Figure 2E*). We observed a band at a size of ~180 KDa, well above the predicted size of 110 KDa, suggesting m18 is post-translationally modified. To evaluate m18 localization we expressed a C-terminal GFP m18 fusion protein in NIH 3T3s and analyzed localization of GFP by fluorescent microscopy. We found that the m18-GFP fusion protein localized mostly to the nucleus (*Figure 2F*). Importantly both m18-HA and m18-GFP constructs still induced expression of RAE-1 similarly to other m18 constructs (*Figure 2—figure supplement 1*).

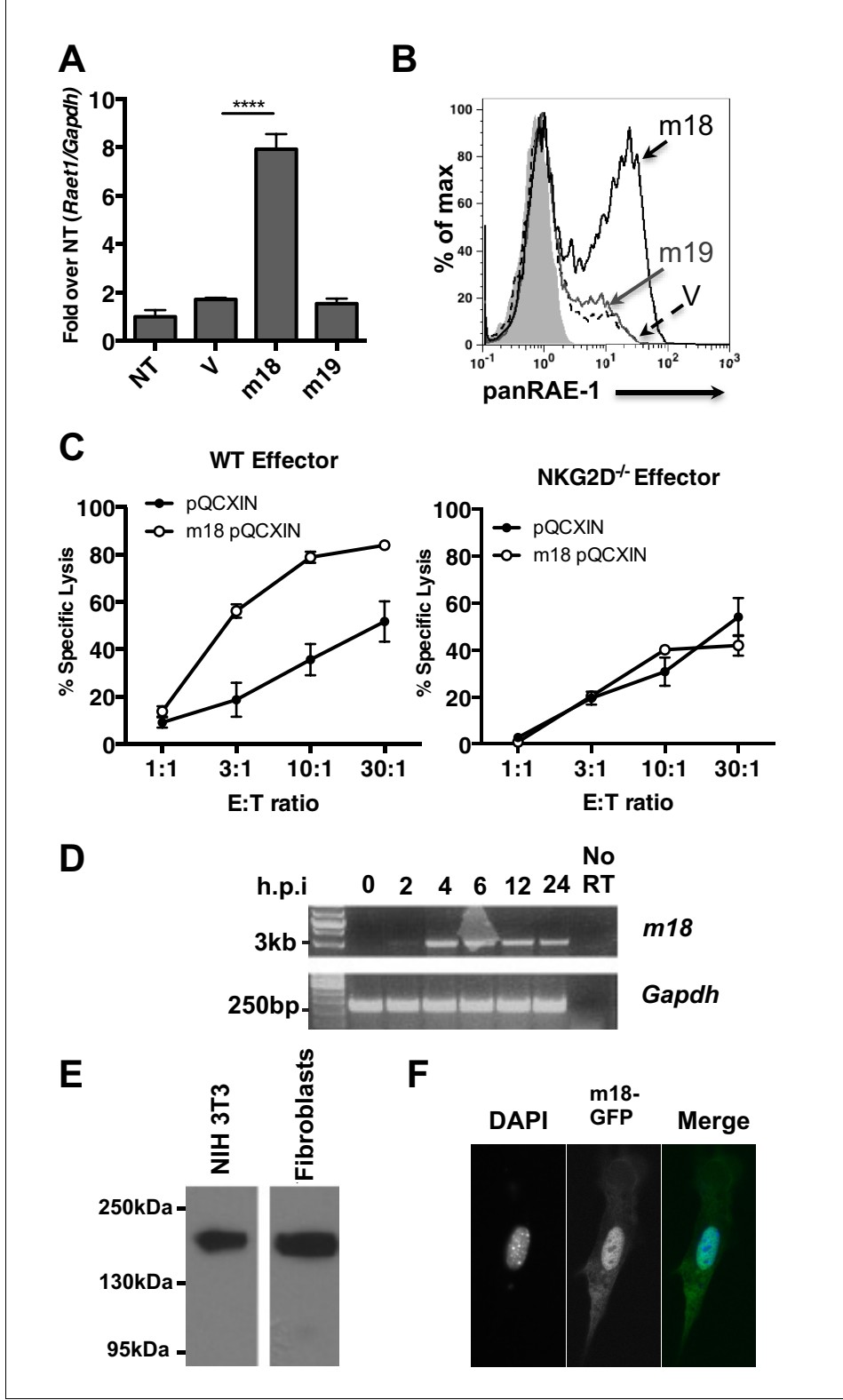

**Figure 2.** m18 expression is sufficient to induce RAE-1 expression and NK cell killing. (**A**) Mouse fibroblasts were transfected with m18, m19, or vector control and analyzed for *Raet1* expression by RT-qPCR Data are normalized to non-transfected fibroblasts (NT) and represented as mean±SEM. Data are representative of three independent experiments. ****p<0.00005 (1 way ANOVA with Bonferroni's multiple comparison post-test). (**B**) Mouse fibroblasts transfected with m18, m19, or vector control plasmid were analyzed for RAE-1 surface expression by flow

*Figure 2 continued on next page*

*Figure 2 continued*

cytometry. Data are representative of three independent experiments. (C) Chromium release assay was performed on fibroblasts transduced with m18 or empty vector control and IL-2 activated NK cells from WT or NKG2D KO b6 littermates. Data are represented as mean ±SEM. Data are representative of three independent experiments. (D) Semi quantitative PCR analysis of m18 expression in fibroblasts infected with MCMV at the indicated times. (E) NIH3T3 or mouse fibroblasts were transfected with an expression plasmid encoding an m18-HA fusion protein, and lysates were analyzed for m18-HA expression by western blot. (F) NIH 3T3 transfected with an expression plasmid encoding an m18-GFP fusion protein, and analyzed for localization of m18-GFP fusion protein by confocal microscopy.

The following figure supplement is available for figure 2:

**Figure supplement 1.** m18 fusion constructs express protein and induce RAE-1 expression.

## The *Raet1e* promoter is activated by m18 through an sp transcription factor binding site

To dissect the mechanism by which m18 induces RAE-1 expression, we measured m18's effect on *Raet1e* promoter activity. An expression vector encoding Firefly luciferase (FLuc) under the control of *Raet1e* promoter (*Jung et al., 2012*) was co-transfected into mouse fibroblasts with a vector encoding m18 or an empty control plasmid. Co-expression of m18 increased the activity of the *Raet1e* promoter relative to vector control (*Figure 3B*). Although E2F sites in the *Raet1e* promoter have been shown to drive RAE-1 expression during proliferation (*Jung et al., 2012*), these sites were dispensable for activation of the *Raet1e* promoter by m18 (*Figure 3—figure supplement 1*).

To identify the promoter elements required for m18 to drive expression from the *Raet1e* promoter, we generated a panel of serial 5′ truncation mutants of the *Raet1e* promoter driving *FLuc* (*Figure 3A*) and co-transfected these with m18 expression vector or control vector. Promoters lacking 15 nucleotides at the 5′ end of the promoter retained WT levels of induction in the presence of m18, but promoter activation by m18 was eliminated when 25 nucleotides were deleted (*Figure 3B*). These data indicate that the *Raet1e* promoter contains an m18 response element (m18RE) between −95 and −85 bp from the transcription start site.

Analysis of the m18RE for transcription factor (TF) binding sites using the JASPAR TF binding database (*Mathelier et al., 2014*) indicated a Specificity factor transcription factor family (Sp TF) binding site (*Figure 3C*) within the m18RE. To determine whether this site was required for m18 to drive expression from the *Raet1e* promoter we mutated this site (m18RE*) (*Figure 3C*) in the context of the rest of the *Raet1e* promoter. Promoter containing m18RE* showed significantly less promoter activity in the presence of m18 than the WT *Raet1e* promoter (*Figure 3D*), suggesting a role for Sp TFs in RAE-1 induction by m18.

We next assessed the ability of Sp TFs to bind the m18RE using an EMSA competition assay. Radiolabeled dsDNA oligonucleotides (oligos) of the m18RE or a Sp consensus binding motif (Sp) were incubated with nuclear lysates in the presence or absence of excess non-radiolabeled oligos of m18RE, m18RE*, Sp, or mutant Sp (that lacks Sp factor binding ie. Sp*). These were then separated on a non-denaturing PAGE by electrophoresis, and the location of the radiolabeled oligo in the gel was measured. Incubation of m18RE or Sp without competing oligo showed high weight shifted bands indicating that these oligos bound factors in the nuclear lysate. The Sp oligo, but not Sp* eliminated the shift of $^{32}$P-m18RE (*Figure 3E*, left panel) indicating that Sp but not Sp* can compete for binding with m18RE. Conversely, the m18RE oligo, but not m18RE*, competed with $^{32}$P-Sp (*Figure 3E*, right panel). Thus, the Sp consensus and m18RE oligos bind the same factors in nuclear lysate, suggesting that Sp TFs regulate *Raet1* transcription.

To assess Sp TF binding to the *Raet1e* promoter, we performed a chromatin immunoprecipitation (ChIP) analysis of Sp TF binding in MCA-205 cells. These cells encode only the *Raet1e* and *Raet1d* isoforms of *Raet1* genes allowing us to circumvent possible complications introduced by the presence of multiple highly homologous isoforms (*Cerwenka et al., 2000*). We first confirmed that MCA-205 cells have low basal RAE-1 expression and that RAE-1 expression is inducible by m18 (*Figure 3—figure supplement 2*). We used Sp1, Sp3, or control IgG antibodies to IP sheared chromatin/DNA complexes from these cells, and quantified the level of *Raet1e* promoter enrichment by

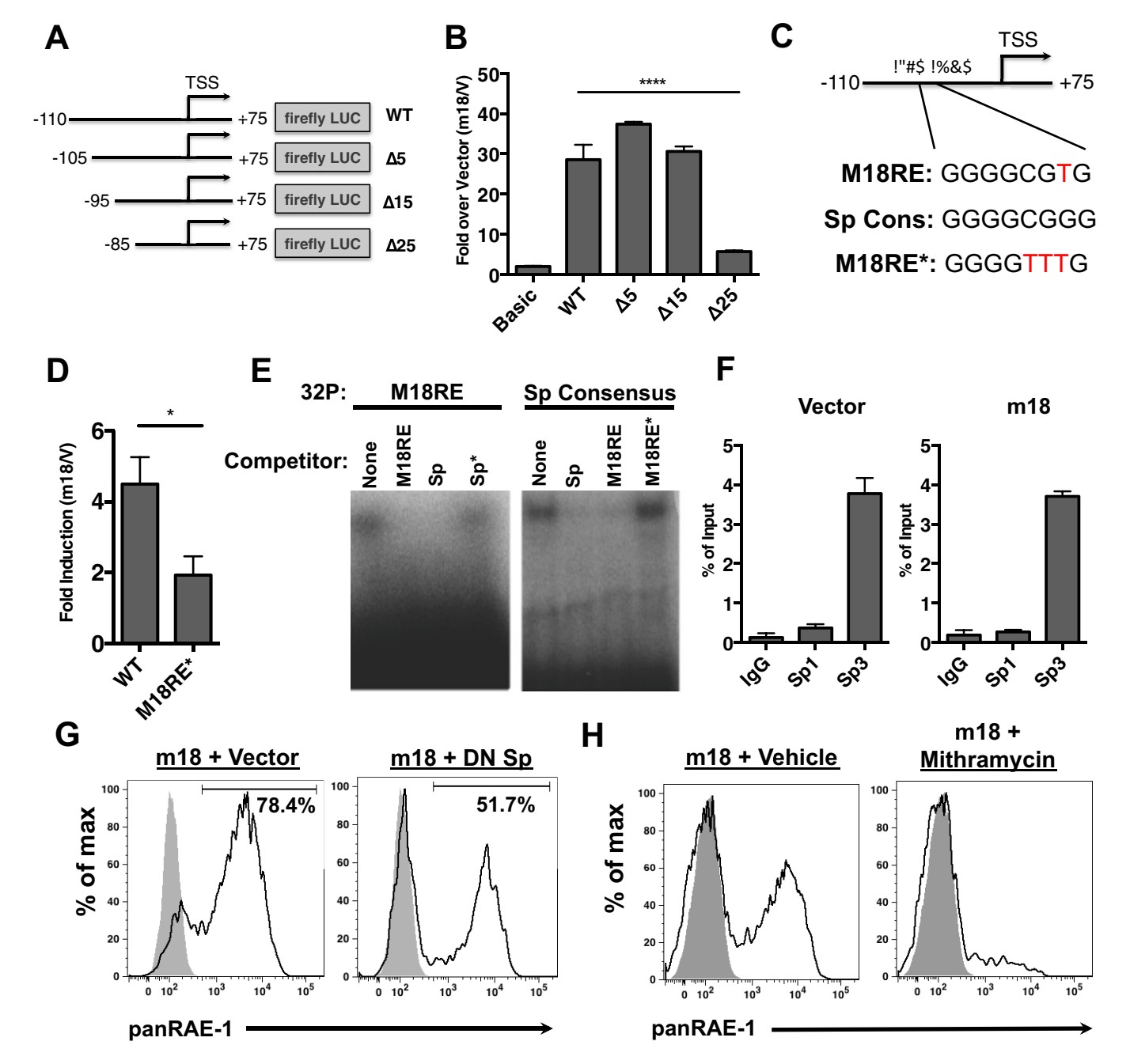

**Figure 3.** m18 activates the *Raet1e* promoter activity through an Sp-transcription factor binding element. (**A**) A graphic representation of Raet1e promoter luciferase constructs used in (**B**). (**B**) *Raet1e* promoter activity was measured by quantifying luminescence in lysates of cells transfected with the indicated luciferase promoter constructs in combination with m18 or vector control. Data are represented as fold increase in luminescence over vector control. Data are represented as mean±SEM. Data are representative of three independent experiments. ****p<0.00005 (1 way ANOVA with Bonferroni's multiple comparison post-test). (**C**) Alignment of the WT m18RE (top), Sp consensus binding sequence (middle) and mutant m18RE (bottom). (**D**) *Raet1e* promoter activity was measured in cells co-transfected with a luciferase construct containing WT *Raet1e* promoter or Raet1e promoter containing a mutation in the Sp-binding site (m18RE*) and either m18 or vector control. Data shows fold increase in luminescence over vector control. Data are represented as mean (±SEM). Data are representative of three independent experiments. *p<0.05 (Student's T-test). (**E**) EMSA was performed on mouse fibroblast nuclear extracts incubated with radio-labeled m18RE (left panel) or radio-labeled Sp consensus sequence (right panel). The indicated non-radiolabeled oligos were added in 1000-fold excess before separation by non-denaturing PAGE. Data is representative of three experiments. (**F**) ChIP was performed on MCA-205 cells using the indicated antibodies and enrichment of the *Raet1e* promoter was assayed by qPCR. Values were normalized to input chromatin. Data are representative of three independent experiments. (**G**) Mouse fibroblasts were co-transfected with m18-RFP and DN-Sp1 GFP or GFP vector control and cells expressing both GFP and RFP were analyzed for RAE-1 expression by flow cytometry. Data is

*Figure 3 continued on next page*

Figure 3 continued

representative of three experiments. (H) Mouse fibroblasts were transfected with m18-GFP and treated with Mithramycin or vehicle control were analyzed for expression of RAE-1 by flow cytometry. Data is representative of three experiments.

The following figure supplements are available for figure 3:

Figure supplement 1. m18 induces transcription from the *Raet1e* promoter independent of E2F binding sites.

Figure supplement 2. MCA-205 carcinoma cell line is inducible for RAE-1 expression by m18.

qPCR. Samples immunoprecipitated using an antibody against Sp3 but not Sp1 showed significant enrichment of the *Raet1e* promoter over the IgG control, indicating that Sp3 occupies the *Raet1e* promoter. Interestingly, m18 expression did not alter the amount of Sp3 bound to the *Raet1e* promoter (*Figure 3G*). This indicates that Sp3 constitutively occupies the *Raet1e* promoter.

To determine whether Sp3 binding is required for m18 induction of RAE-1 expression, we co-transfected mouse fibroblasts with expression vectors encoding m18 and a dominant negative Sp TF (DN-Sp) and measured RAE-1 surface expression levels by flow cytometry. DN-Sp consists of an Sp1 protein DNA binding domain lacking the transactivation domain. DN-Sp binds to Sp1/3 binding sites and acts as a competitive inhibitor of promoter activation (*Won et al., 2002*). Co-expression of DN-Sp and m18 resulted in a decreased percentage of RAE-1 expressing cells (*Figure 3G*) suggesting that Sp factors are important for m18 to drive RAE-1 expression. To further test this hypothesis we transfected mouse fibroblasts with an expression vector encoding m18 and treated these cells with mithramycin, an inhibitor of Sp factor binding to DNA (*Blume et al., 1991*). Mithramycin treatment reduced RAE-1 induction by m18 (*Figure 3H*) further indicating that Sp TF activity is required for RAE-1 induction by m18.

## The *Raet1e* promoter is repressed by HDAC3 in an Sp-dependent manner

The human NKG2D ligand ULBP-1 is repressed by histone deacetylase 3 (HDAC3) in the absence of stress. HDAC3 is recruited to the ULBP1 promoter by Sp3 (*López-Soto et al., 2009*). Additionally, HDAC inhibition induces RAE-1 expression (*Gasser et al., 2005*). Consistent with these findings, we observed that the chemical HDAC inhibitor butyrate activated the WT *Raet1e* promoter (*Figure 4A*). Interestingly, butyrate treatment failed to drive expression from the m18RE* mutant *Raet1e* promoter (*Figure 4A*). To assess whether chemical HDAC inhibition also requires Sp TFs to drive RAE-1 expression we treated mouse fibroblasts with butyrate in combination with mithramycin. Mithramycin treatment reduced RAE-1 expression in response to butyrate treatment (*Figure 4B*), indicating that Sp TFs are also required to drive RAE-1 expression in response to HDAC inhibition.

Given that histone deacetylase inhibitors induce the expression of RAE-1 we wanted to identify which HDAC family member(s) must be inhibited to drive RAE-1 expression. To identify specific HDAC inhibitors that induce RAE-1 expression, cells were treated with a panel of HDAC inhibitors and analyzed for RAE-1 expression by flow cytometry. The pan-HDAC inhibitors TSA and NaB induced RAE-1 expression, as did the HDAC1/3 inhibitor MS-275 and the HDAC3 inhibitor RGFP966. In contrast, an HDAC1 inhibitor (4-(dimethylamino)-N-[6-(hydroxyamino)−6-oxohexyl]-benzamide), and an HDAC6,8 inhibitor (droxinostat) did not (*Figure 4C*). These results indicate that HDAC3 is involved in RAE-1 repression, likely through recruitment to the promoter by Sp3 (*López-Soto et al., 2009*) though additional HDACs may be involved.

## m18 expression increases histone acetylation

As m18 and chemical HDAC inhibitors drive expression through the same promoter element in the *Raet1* promoter, we hypothesized that m18 acts as an HDAC inhibitor. HDACs modulate gene expression by deacetylating histones to maintain closed chromatin and repress gene expression. Thus one major prediction of this hypothesis is that histone acetylation should be increased in cells expressing m18.

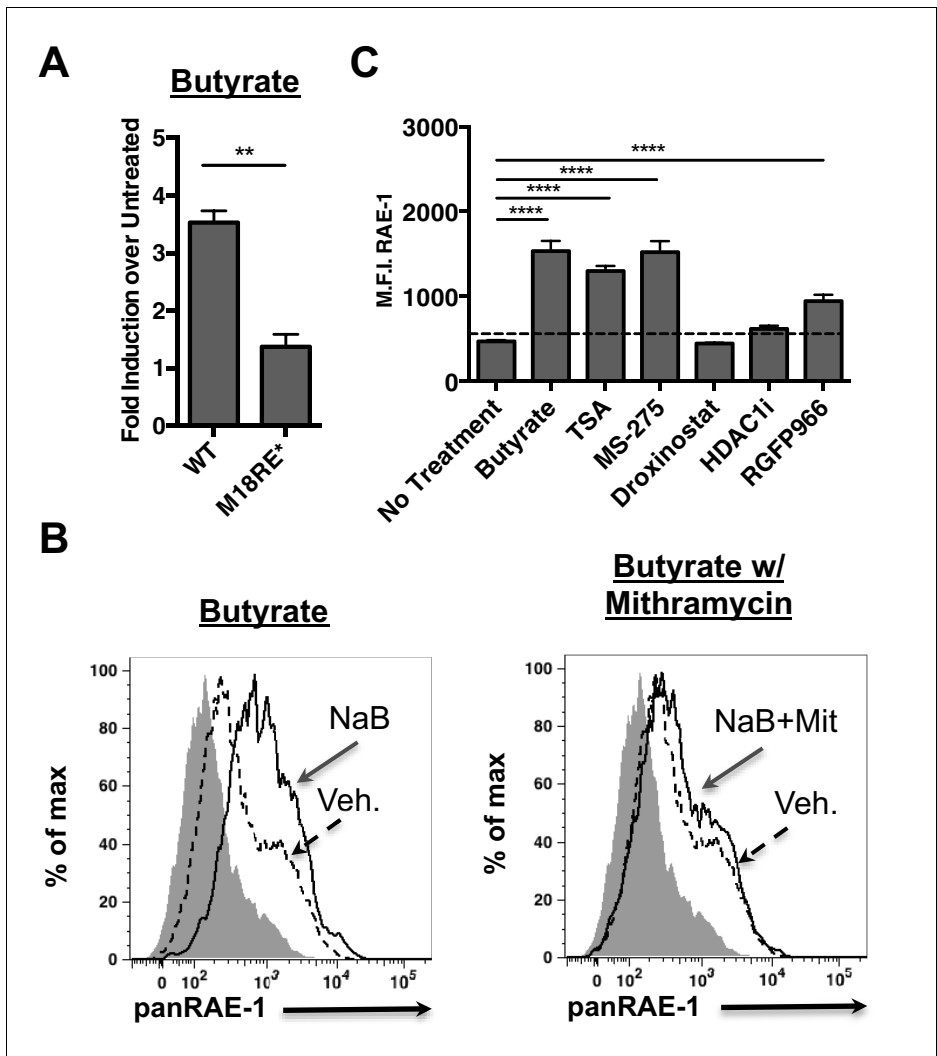

**Figure 4.** HDAC inhibition drives *RAE-1* expression in an Sp factor dependent manner. (**A**) *Raet1*e promoter activity was measured in lysates from mouse fibroblasts transfected with either WT *Raet1e* promoter or the m18RE* promoter treated with sodium butyrate (NaB) (1 mM). Data are expressed as fold change between butyrate treated and untreated promoter. Data are represented as mean±SEM. Data are representative of three independent experiments. **p<0.005 (Student's T-test). (**B**) Cells treated with NaB (1 mM) with or without Mithramycin (1.5 µM) were analyzed for RAE-1 expression by flow cytometry. Data are representative of five independent experiments. (**C**) Cells were treated with HDAC inhibitors TSA (1 nM) (pan-HDACi), NaB (0.1 mM) (Class I and IIa), MS-275 (800 nM) (HDAC1,3), RGFP966 (640 nM) (HDAC3) 4-(dimethylamino)-N-[6-(hydroxyamino)− 6-oxohexyl]-benzamide (HDAC1i) (1 µM) (HDAC1), or Droxinostat (3 µM) (HDAC6,8) and analyzed for RAE-1 expression by flow cytometry. Data are represented as mean fluorescent intensity±SEM. Data are representative of three independent experiments. ****p<0.00005 (1 way ANOVA with Bonferroni's multiple comparison post-test).

To test the hypothesis that m18 expression increases histone acetylation we analyzed lysates of mouse fibroblasts transduced with m18 or vector control for acetylated histone 3 (AcH3) as well as bulk histone 3 (H3) by western blot. AcH3 levels were increased in m18 expressing mouse fibroblasts, as compared to those expressing vector control. H3 levels were unchanged (*Figure 5A*). To determine whether this effect was taking place as a direct result of m18 expression we transfected mouse fibroblasts with a construct encoding m18 C-terminally fused to red fluorescent protein (RFP), and assessed AcH3 levels by immunofluorescence assay (*Figure 5B*). We compared the fluorescent intensity of AcH3 staining in the nucleus of m18 expressing cells, to non-transfected cells in the same field of view, and found significantly higher levels of AcH3 in m18 expressing cells. AcH3

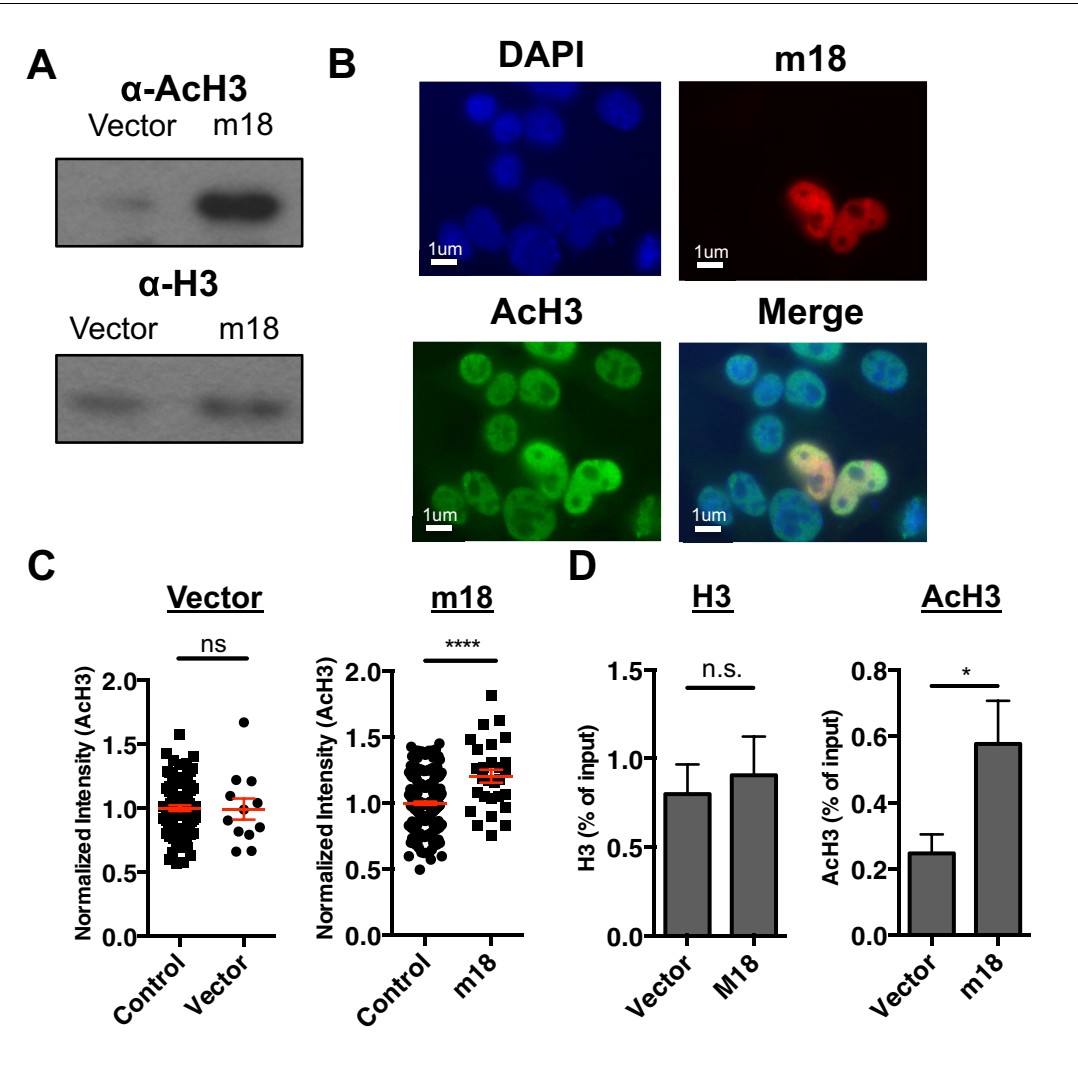

**Figure 5.** m18 expression increases levels of histone acetylation. (**A**) Mouse fibroblasts stably expressing m18 or vector control were analyzed for H3 and AcH3 expression by western blot. Data are representative of three independent experiments. (**B**) Representative image of cells transiently transfected with vector encoding m18 with a C-terminal fusion to RFP (m18-RFP) and stained for AcH3. (**C**) Quantification of AcH3 levels in cells expressing m18-RFP or RFP control vector from compared to non-transfected controls in same field of view. Red bars are representative of mean±SEM. Data are representative of three independent experiments. ****p<0.00005. n.s., not significant. (Student's T-test). (**D**) ChIP was performed for AcH3 and H3 in MCA-205 mouse carcinoma cells stably expressing m18 or vector control. Data are normalized to input chromatin and represented as mean±SEM. Data are representative of three independent experiments. *p<0.05, n.s., not significant (Student's t-test).

The following figure supplement is available for figure 5:

**Figure supplement 1.** m18 expression does not change Histone three levels.

level was unchanged in cells transfected with RFP alone (*Figure 5C*). Expression of m18-RFP or RFP alone did not affect H3 levels (*Figure 5—figure supplement 1*). Finally, we assessed whether m18 expression increased acetylation at the *Raet1e* promoter by performing ChIP against AcH3 or H3 out of lysates from MCA-205 cells transduced with m18 or vector control. Analysis of *Raet1e* promoter enrichment showed an increase in AcH3 levels associated with the *Raet1e* promoter in m18 expressing cells while H3 levels were unchanged (*Figure 5D*). Together, these results indicate that m18 increases histone acetylation in cells, including at the *Raet1e* promoter.

## CK2 inhibition connects m18 to HDAC3 regulation

To investigate the mechanism by which m18 increases histone acetylation, we analyzed m18 binding to host proteins by immunoprecipitation (IP) followed by mass spectrometry. N- or C-terminally Strep-tagged m18 was transiently expressed in HEK293T cells, and native m18 complexes were captured by streptactin-affinity capture. Proteins were identified with peptide sequencing by LC-MS/MS, and the results are reported in *Supplementary files 1* and *2*. We used HEK293T for these assays as they allowed for direct comparison with similar virus-host AP-MS samples, and enabled us to identify high frequency background proteins found in this system (*Supplementary file 2*) (previously reported in 30). To rank the specificity of the remaining candidate protein-protein interactions, a *Z*-score was calculated with three affinity experiments and 598 unrelated virus-host APMS experiments (*Supplementary file 3*). This allowed identification of the most specific interaction as casein kinase II beta (CSNK2B) in APMS samples from human HEK293T cells. Peptide sequencing data from LC-MS/MS analysis covered 48–55% of the annotated m18 ORF protein sequence overall, and included phosphorylations at several S/T sites as well as peptides from the Strep-tag (*Supplementary file 4*). We confirmed the interaction with CK2 in NIH-3T3 by IP-Western blot (*Figure 6A*).

CK2 is involved in many biological processes, including DNA damage signaling (*Ghavidel and Schultz, 2001*), apoptosis (*Hellwig et al., 2010*), and cell cycle progression (*Homma and Homma, 2008*). CK2 is a constitutively active kinase that phosphorylates HDAC3 at serine 424 (S424) activating it (*Zhang et al., 2005*). The activation of HDAC3 by CK2 has previously been demonstrated to help drive chromatin condensation during mitosis (*Patil et al., 2016*) and has been shown to repress expression of genes during beige fat thermogenesis (*Shinoda et al., 2015*) indicating that this interaction can functionally regulate both chromatin structure and gene expression. Thus, we hypothesized that CK2 activation of HDAC3 represses *Raet1* transcription in healthy cells, while inhibition of CK2 by m18 would result in less HDAC3 activity and thus induce *Raet1* transcription.

To test this hypothesis we investigated the ability of CK2 specific inhibitors to induce the expression of RAE-1. As CK2 acts as an anti-apoptotic factor (*Hellwig et al., 2010*) long-term inhibition of CK2 leads to apoptosis (*Ruzzene et al., 2002*). To circumvent this technical challenge, mouse fibroblasts were treated with zVAD caspase inhibitor in combination with the chemical inhibitor of CK2, TBBt, and cell surface RAE-1 expression was analyzed by flow cytometry. TBBt treatment induced RAE-1 expression in excess of zVAD alone (*Figure 6D*), indicating CK2 is a negative regulator of RAE-1 expression.

CK2 activates HDAC3 by phosphorylating HDAC3 at S424, thus CK2 inhibition by m18 should reduce levels of HDAC3 S424. To test this prediction we analyzed the HDAC3 S424 phosphorylation status in mouse fibroblasts expressing m18. We transfected mouse fibroblasts with a construct encoding m18 C-terminally fused to RFP or a control vector only expressing RFP, and assessed HDAC3 S424 levels by IFA (*Figure 6E*). We compared the fluorescent intensity of HDAC3 S424 staining in the nucleus of m18 expressing cells, to non-transfected cells in the same field of view, and found significantly higher levels of HDAC3 S424 in m18 expressing cells (*Figure 6F*). By the same measure overall levels of HDAC3 were unaffected by m18 transfection (*Figure 6—figure supplement 1*). This decrease was not observed in mouse fibroblasts transfected with control RFP vector. Together, these results indicate that m18 inhibits HDAC3 activity by binding to CK2 and reducing HDAC3 phosphorylation.

## m18 is not required for MCMV growth in vitro or acutely in vivo

The fact that m18 ORF is highly conserved across strains of MCMV (*Khairallah et al., 2015*) suggests that m18 exert an essential function in MCMV life cycle. Such function is unlikely to be ligand upregulation as their expression would be detrimental for viral fitness (and accordingly the virus actively evades them). To assess whether m18 was required for growth in vitro we infected NIH 3T3 cells with WT or Δ18 MCMV and assessed viral output by plaque assay 2,3,4,5 and six days post infection. Δ18 MCMV did not grow significantly different from WT MCMV in this assay (*Figure 7A*).

We then sought to determine whether m18 is critical for viral growth in vivo. As the Δ18 MCMV includes a large (3 kB) genome deletion we generated an additional mutant MCMV to eliminate m18 expression with minimal alteration to the rest of the genome. This virus mutant (MCMV[m18stop]) includes a set of two stop codons early in the m18 coding sequence preventing the production of

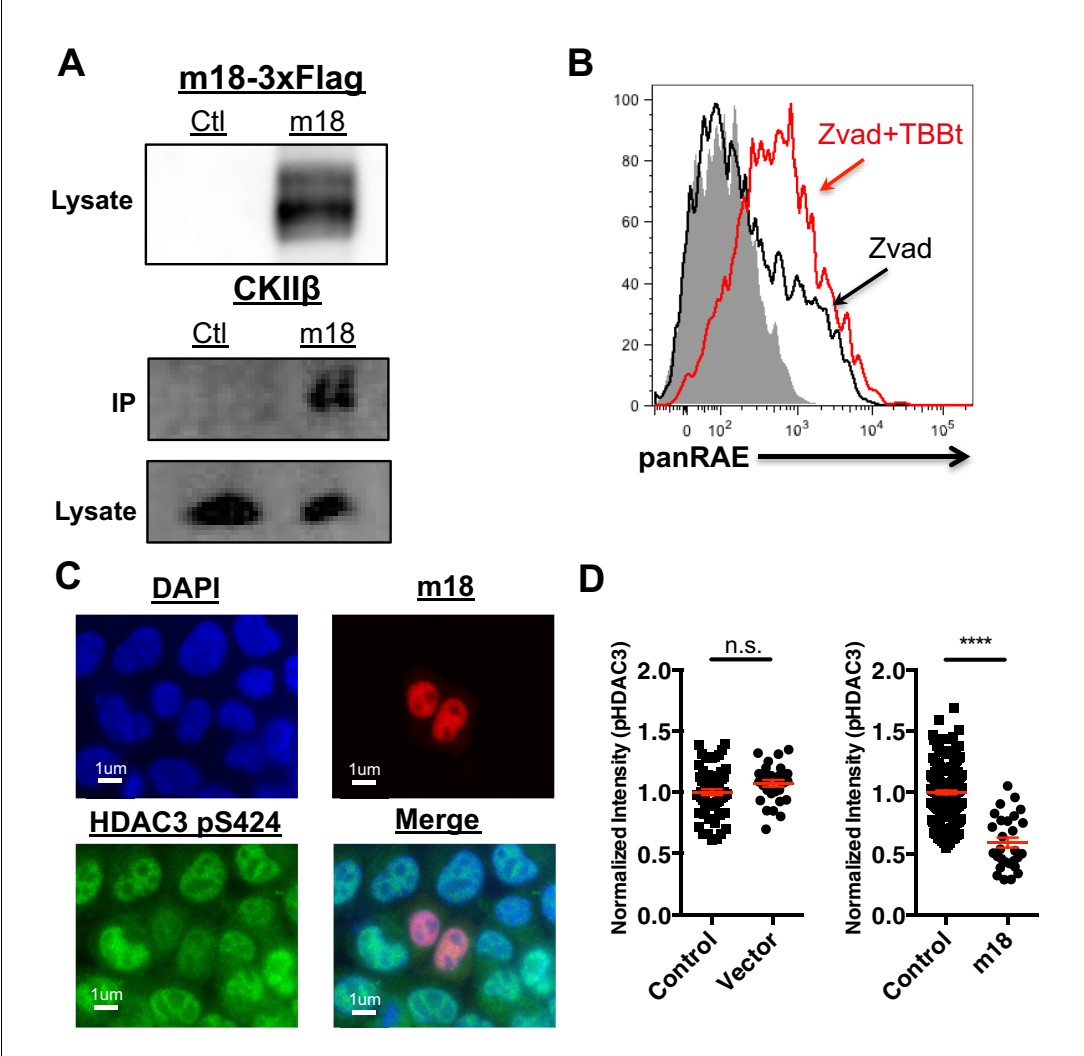

**Figure 6.** CK2 directly interacts with m18 and represses HDAC function. (**A**) Immunoprecipitation (IP) of m18 was performed in lysates of cells expressing m18-3xFlag or empty 3xFlag vector, and the products were analyzed for FLAG and CK2β by western blot. Data are representative of three independent experiments. (**B**) Mouse fibroblasts were treated with CK2 inhibitor TBBt in conjunction with zVAD or zVAD alone and analyzed for RAE-1 expression by flow cytometry. Data is representative of three independent experiments. (**C**) Representative image of fibroblasts expressing m18-RFP and stained for HDAC3 pS424. (**D**) Quantification of HDAC3 pS424 levels in cells expressing m18-RFP or RFP control vector from compared to non-transfected controls in same field of view. Red bars are representative of mean±SEM. Data are representative of three independent experiments. ****p<0.00005. n.s., not significant. (Student's T-test).

The following figure supplement is available for figure 6:

**Figure supplement 1.** m18 expression does not change HDAC3 levels.

the m18 protein. Like Δ18 MCMV, MCMV[m18stop] fails to induce RAE-1 expression in mouse fibroblasts, but has no defect in IE-1 expression (*Figure 7—figure supplement 1*) and growth was similar to wild type virus.

To assess the roll of m18 in vivo we infected BALB/c mice by intraperitoneal (i.p.) injection with WT MCMV or MCMV[m18stop], and assessed the levels MCMV five days later in the spleen, liver and lungs of these mice by qPCR (*Smith et al., 2008*). Mice infected with WT or MCMV[m18stop] had identical levels of MCMV in all organs assayed at this time (*Figure 7B*). This indicates that m18 elimination has no impact on viral growth in WT mice.

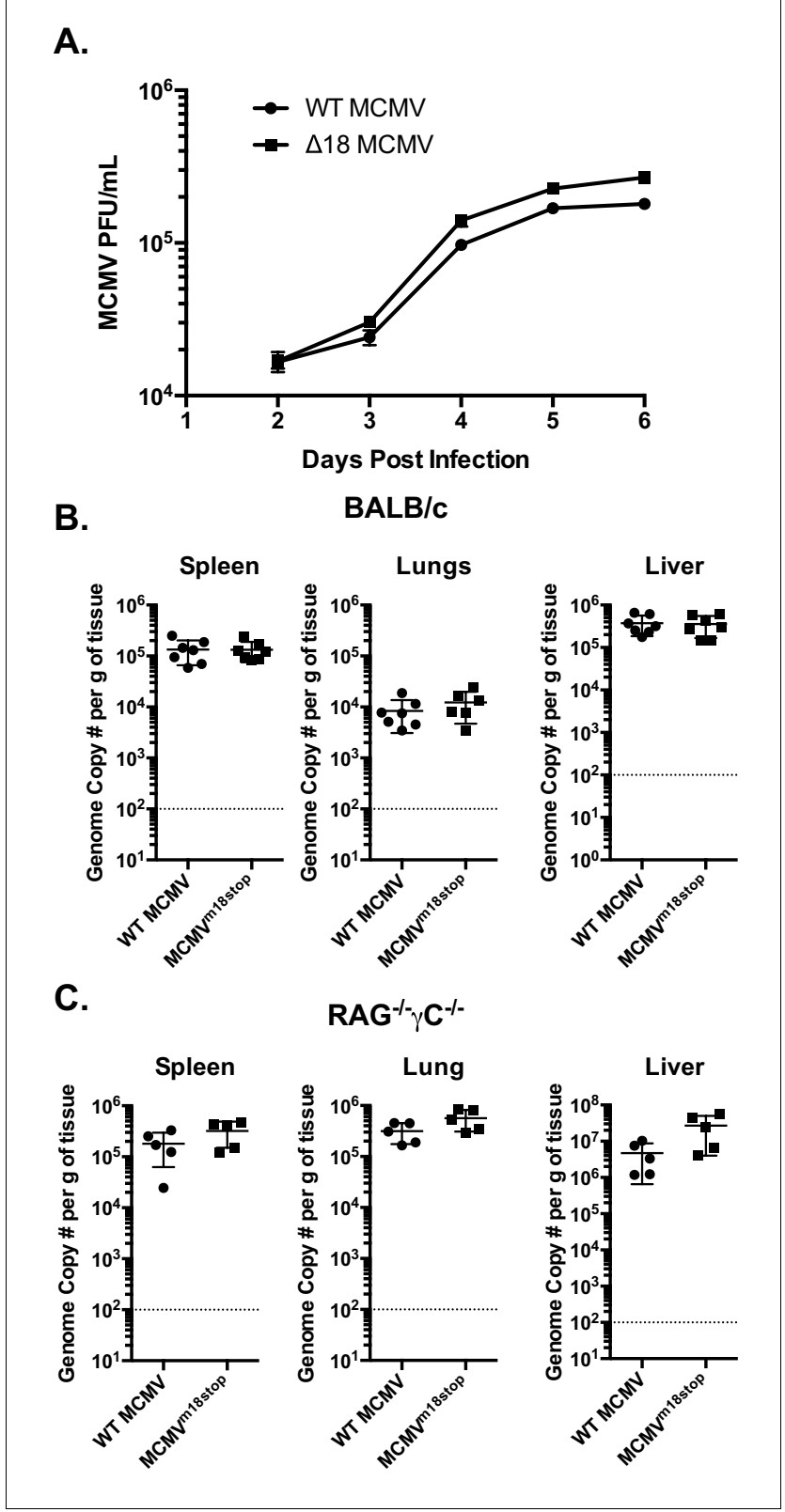

**Figure 7.** m18 is dispensable for MCMV growth in vitro and acutely in vivo. (A) MCMV titers in supernatents of NIH-3T3 cells infected with either WT or Δ18 MCMV. (B) qPCR analysis of MCMV genome copy number from spleen, lung, and liver tissues of BALB/c mice infected with 500,000 p.f.u. of WT or MCMV[m18stop]. (C) qPCR

*Figure 7 continued on next page*

*Figure 7 continued*
analysis of MCMV genome copy number from spleen, lung, and liver tissues of RAG$^{-/-}$γC$^{-/-}$ Mice infected with 50,000 p.f.u. of WT or MCMV$^{m18stop}$.
The following figure supplement is available for figure 7:

**Figure supplement 1.** MCMV$^{m18stop}$ does not induce RAE-1 expression or cause a defect in IE-1 expression.

It is possible that a deleterious effect of losing m18 was masked in WT mice by immune surveillance. To assess viral fitness in the absence of immune surveillance, we assessed the growth of WT MCMV and MCMV$^{m18stop}$ in RAG$^{-/-}$ yC$^{-/-}$ mice. These mice lack T cells, B cells, and NK cells. Interestingly, these mice also showed identical levels of MCMV between the WT and mutant viruses in spleen, lung, and liver (*Figure 7C*). Indicating that m18 is dispensable for viral growth at this early time point.

## Model for m18 induced RAE-1 expression in MCMV infection

Together our results suggest a model for RAE-1 regulation in which HDAC3, constitutively activated by CK2, maintains *Raet1* in a repressed state. During MCMV infection m18 protein interacts with CK2 and prevents the activation of HDAC3, reducing its repressive activity. The *Raet1* promoter becomes unrepressed and constitutively bound Sp3 can recruit transcriptional machinery to transcribe *Raet1*. To circumvent this induction that would target infected cells for elimination by NK-Cells, MCMV encodes a number of highly efficient evasins that prevent NKG2D from recognizing RAE-1 during viral infection (*Figure 8*).

## HDAC inhibitors of other herpesviruses induce expression of NKG2D ligands

Some other herpesviruses also encode viral HDAC inhibitors. Two prominent examples of HDAC inhibiting viral proteins are IE1 from HCMV (*Nevels et al., 2004*) and ICP0 from HSV-1 (*Gu et al., 2005*). We sought to determine whether expression of these proteins would be sufficient to induce expression of human NKG2D ligands. We transfected human foreskin fibroblasts (HFFs) with plasmids expressing IE1 or ICP0 or empty vector and measured expression of human NKG2D-ligands by RT-qPCR compared to vector control. Both ICP0 and IE1 induced expression of human ULBP1 (*Figure 9A, B*). These results suggest that viral inhibition of HDACs is a common mechanism driving NKG2D ligand induction in humans and mice.

## Discussion

Our results suggest a model in which RAE-1 expression is repressed in the absence of stress signals due to closed chromatin around the *Raet1* locus. This repression is maintained by constitutive phosphorylation and activation of HDAC3 by CK2. However, during MCMV infection, m18 reduces phosphorylation of HDAC3 by directly interacting with CK2 an activator of HDAC3. As a result, HDAC3 becomes less activated and the chromatin around the *Raet1* locus becomes acetylated and accessible. Sp3 is then able to recruit the general transcriptional machinery to transcribe *Raet1* (*Figure 8*). These findings reveal how the chromatin environment contributes to silencing of *Raet1* in unstressed cells, as well as how a single viral protein relieves this repression.

It is notable that HDAC inhibition is a feature common to a number of herpesviruses. Many of the known HDAC inhibiting proteins from herpesviruses are crucial to viral fitness (*Nevels et al., 2004*; *Gu et al., 2005*). Our work demonstrated that for viruses, encoding HDAC inhibiting proteins comes at the cost of inducing NKG2D ligand transcription. It is thus tempting to speculate that viruses being unable to replicate efficiently without inhibiting HDACs have had to evolve a plethora of proteins that reduce NKG2D ligand expression at the protein level (*Jonjić et al., 2008*).

While it is appreciated that viral HDAC inhibitors can improve viral fitness, the precise significance of virally encoded HDAC inhibitors in herpesvirus pathogenesis is still not well understood. Cellular HDAC enzymes have been shown to repress viral gene expression and, in some cases replication. In the context of lytic infection herpes viruses must overcome this repression using viral HDAC

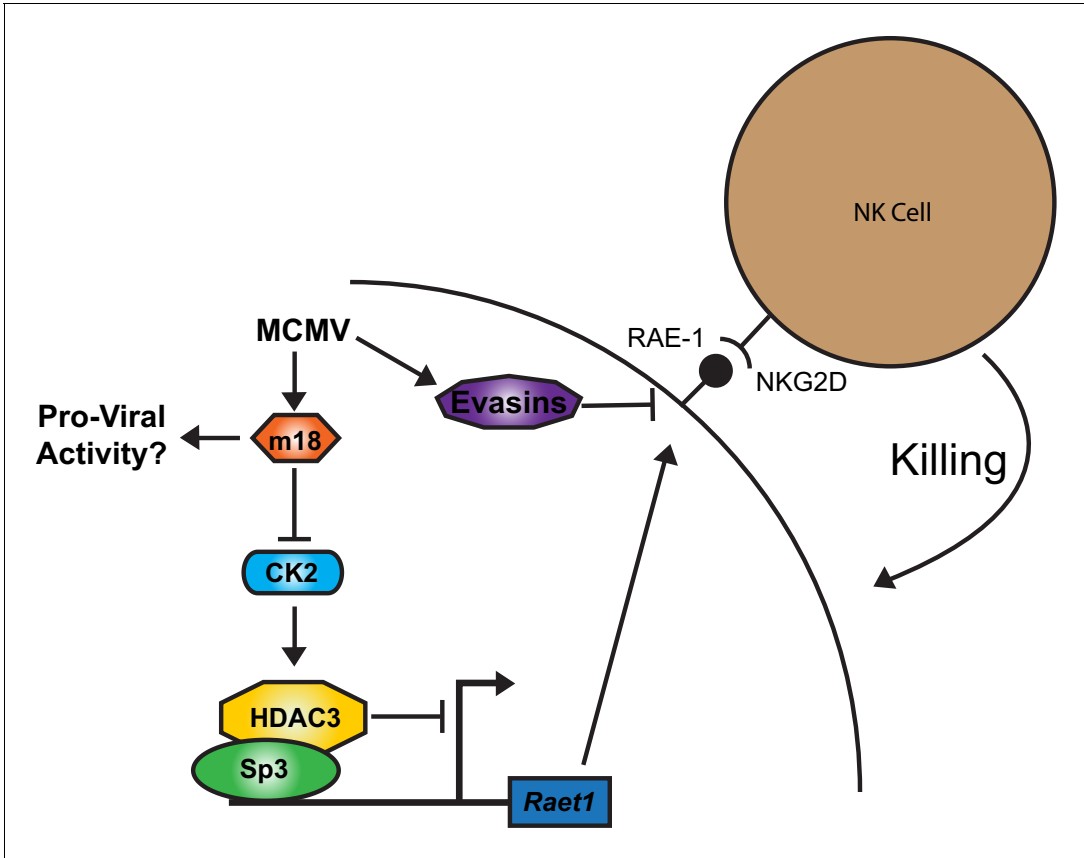

**Figure 8.** A model for RAE-1 induction by m18 during MCMV infection. In the absence of stimulus CK2 phosphoylates and activates HDAC3, which represses the *Raet1* promoter. During MCMV infection m18 directly interacts with CK2 preventing activation of HDAC3 and allowing the *Raet1* promoter to become de-repressed, and permiting the expression of *Raet1*. MCMV also encodes a variety of evasins that prevent the cell surface expression of RAE-1 in order to evade recognition and killing by NK cells. Together these systems allow the virus to inhibit HDAC3 activity, while evading the deleterious effects of inducing NKG2D ligands.

inhibitors (*Nevels et al., 2004*), in fact chemical HDAC inhibition can rescue a defect in viral replication caused by HCMV IE1 deficiency (*Nevels et al., 2004*). Additionally, viral gene repression by HDACs also promotes viral latency. HDACs have been shown to occupy the promoters of the immediate early genes that drive reactivation in HCMV and MCMV (*Murphy et al., 2002*; *Liu et al., 2010*), and have been shown to repress KSHV reactivation (*Shin et al., 2014*). In fact, inhibition of HDACs is sufficient to drive reactivation in latently infected KSHV cell lines (*Miller et al., 1997*). In these cases viral HDAC inhibitors may provide a way for the virus to promote its own reactivation from latency. Importantly, these proteins have been described in human viruses and their function in vivo has not been well established. We do not observe a role for m18 early in MCMV infection in vivo but it is possible that m18 inhibits HDACs in order to benefit the virus during later stages of infection such as during establishment of MCMV latency or reactivation from latency similar to KSHV. Thus, it will be interesting in the future to determine if m18 has alternative functions that contribute to viral fitness during later stages of infection.

Our data provide the first evidence for the role of CK2 in NKG2D regulation. Given that CK2 activity can be modulated by many stress pathways, it will be of interest to assess whether CK2 contributes to NKG2D ligand regulation in situations such as DNA damage. Furthermore, CK2 is broadly involved in many cellular processes. Thus m18 inhibition of CK2 is likely to have additional effects on the cell. Chemical CK2 inhibitors enhance the anti-viral effect of Type I interferon signaling during HSV-1 infection (*Smith et al., 2011*), implicating CK2 as a key factor in host response to viral

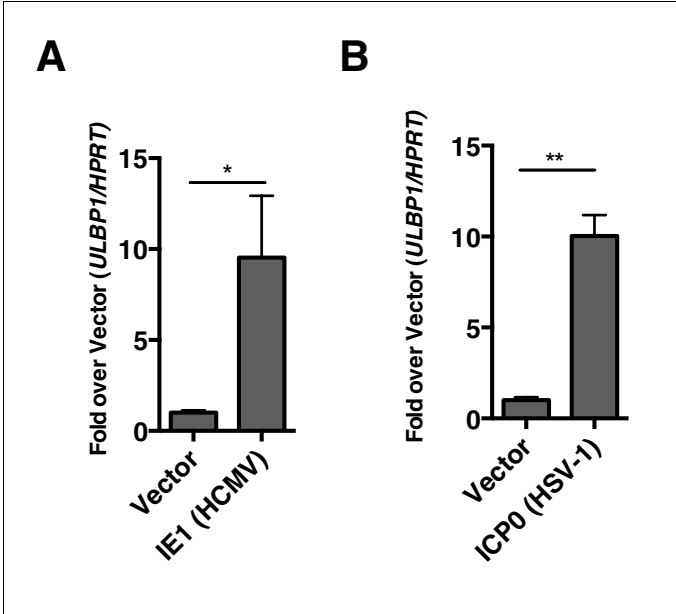

**Figure 9.** Virally encoded HDAC inhibitors from human herpesviruses induce human NKG2D ligand expression. (A, B) RT-qPCR analysis of *ULBP1* expression in Human foreskin fibroblasts (HFFs) transfected with empty vector or IE-1 (HCMV), or ICP0 (HSV-1). Data are normalized to vector control and represented as mean±SEM. Data are representative of three independent experiments.

infection. It will be interesting in the future to elucidate how manipulation of CK2 by m18 impacts the host during viral infection.

In addition to their role in viral infection, HDACs are also important for the control of cancer. Expression of Class I HDAC enzymes (including HDAC3) is increased in cancers (*Glozak and Seto, 2007*). Increased HDAC expression promotes carcinogenesis through down regulation of tumor suppressors such as p21 in an Sp1/Sp3 dependent manner (*Wilson et al., 2010*). As NKG2D ligand ULBP-1 is suppressed by HDAC3, it has been proposed that increased HDAC expression also contributes cancer's ability to evade NKG2D recognition (*López-Soto et al., 2009*). Several chemical HDAC inhibitors, including some used in this study, are being developed as anti-cancer drugs (*West and Johnstone, 2014*). While these drugs act in multiple ways, one effect may be to induce expression of NKG2D ligands leading to increased NK cell recognition and NK cell activation (*López-Soto et al., 2009*; *West and Johnstone, 2014*). Our study suggests that HDAC3-specific inhibitors may be particularly adept at leveraging this aspect of anti-cancer activity. CK2 is also highly upregulated in cancers (*González et al., 2008*), and one intriguing possibility is that this may contribute to NKG2D ligand repression and NK cell evasion.

The regulation of NKG2D ligands is an important pivot point in immune regulation. Active repression of NKG2D ligand transcription by HDACs provides a way for the host to repress NKG2D ligand expression in the absence of stress while allowing for the possibility of expression when experiencing stress. Our study suggests that this system may also provide a strategy for hosts to recognize and respond to viral infection.

## Materials and methods

### Cell lines and reagents

All cells cultured in DMEM with 10% FBS (Invitrogen, Carlsbad CA) and 100 μ/mL Penicillin/Streptomycin (Invitrogen) unless otherwise noted. NIH 3T3 cells (ATC#CRL-1658) were obtained from the ATTC. HFF-1 cells (ATCC#SCRC-1041) were obtained from the ATTC. Mouse fibroblasts were generated as described (*Gasser et al., 2005*) and provided by Pr D Raulet. MCA-205 were received from

Pr. L. Linear (UCSF). All cell lines tested negative for mycoplasma as described by *Young et al. (2010)*. Mithramycin A and Butyrate were purchased from Sigma (St. Louis, MO). RGFP966 was purchased from Seleckchem (Houston, TX). Antibodies recognizing Sp1 (product # 07–645), Acetylated Histone 3 (product # 06–599) were purchased from Millipore (Billerica, MA). Antibodies against histone 3 (clone D1H2), HDAC3 (Clone 7G685), and antibody recognizing CKII substrate (#8738) were purchased from Cell Signaling Technologies (Danvers MA). Antibody recognizing CK2β (Product PA5-27416) was purchased from Thermo Fisher (Waltham, MA). Antibody against Sp3 (D-20) was purchased from Santa Cruz Biotechnology. Plasmid encoding HSV-1 ICP0 was provided by the Knipe Lab (Harvard). Plasmid encoding HCMV IE1 was provided by the Weinberger Lab (UCSF).

## Transfection

Cells were transfected using FuGENE HD reagent (Promega, Madison WI) according to manufacturer's instructions.

## RT-qPCR and qPCR

RNA from mouse fibroblasts or human foreskin fibroblasts was extracted in Trizol (Invitrogen). DNA was removed through treatment with RQ1 DNase (Promega), RNA abundance and quality was measured using a nanodrop ND-1000 to analyze 260/230 ratios. and 1 µg of RNA was reverse transcribed for 45 min at 42°C using oligo(dT) primer (IDT) and SuperScript II (Invitrogen) in 20 µl total volume. cDNA was analyzed using an ABI7300 RT-qPCR System and cycled using a 95°C dissociation step for 15 s and a 60°C amplification step for 1 min for 40 cycles. Samples were prepared as 1 µl of prepared cDNA with 10 µl of iTAQ universal Syber Green supermix (Invitrogen) with primers at a concentration of 300 nM in a total reaction volume of 20 µl. Cq values were determined using the Applied Biosystems 7300 SDS software. All samples were run as triplicates from the same pool of cDNA and the results averaged. Average Cq values were then normalized by ΔΔCT against the indicated reference gene. Biological replicates were then used to calculate mean and standard deviation of values. Between 3 and 5 biological replicates were used in each experiment. Samples without RT were included to control for DNA contamination. RAE-1 primers were described previously (*Tokuyama et al., 2011*), ULBP-1 primers can be found in *Table 1*.

For DNA qPCR DNA was extracted from single cell suspensions of mouse tissue using Quiagen DNeasy Blood and Tissue Kit (Qiagen). DNA abundance and quality was measured using a nanodrop ND-100 to analyze 260/290 ratios. Samples for use in qPCR were prepared as 2 µL of isolated DNA with 10 µL of iTAQ universal Syber Green supermix (Invitrogen) with primers at a concentration of 300 nM, and cycled as described above with Cq values calculated as detailed above. Between 5–7 biological replicates were used in each experiment, tissues from uninfected mice were included to control for viral DNA contamination, and buffer processed without tissues were included to control for non-viral DNA contamination. MCMV gB primers were described previously by Khairallah and colleges. Standard curve for the calculation of absolute genome number was done using known quantity of purified MCMV BAC. Genome copy per gram tissue was calculated from weight of starting material and genome copies in each sample. Limit of detection was defined for each experiment and tissue as the copy number calculated from the average Cq from uninfected tissue.

## Virus production, propagation, and infection

Mutant MCMV lacking genes m01 through m22 (MCMVΔ1–22) was a gift from Dr. Hidde Ploegh (Whitehead Institute, MIT, MA). E. coli strain DH10B were transformed with MCMV BAC pSM3fr and a plasmid encoding the arabinose-inducible Red recombination genes, pkD46, were obtained from Dr. Martin Messerle (Hannover Medical School, Germany). Kanamycin cassette was amplified from

**Table 1.** qPCR primers.

| Gene symbol | Accession number | Forward | Reverse | Amplicon length | Location |
|---|---|---|---|---|---|
| *ULBP1* | Q9BZM6 | gccaggatgtcttgtgagcatgaa | ttcttggctccaggatgaagtgct | 134 | Exon 3 |
| *HPRT* | P00492 | ggtgaaaaggaccccacgaag | ggactccagatgtttccaaac | 205 | Exon Spanning (7–9) |

pACYC177 (NEB, Ipswitch MA) containing 50 bp sequences on both ends homologous to the region of interest (*Table 2*). Transformation and induction of recombination was performed as described (*Young et al., 2010*). E. coli strain GS1783 containing MCMV pSM3fr was provided by Dr. Caroline Kulesza (Fort Lewis College) and used to perform scarless BAC recombination as described by Tischer and colleagues (*Borst et al., 2007*). The resulting BAC products were analyzed for anticipated mutation by PCR and EcoRI digestion. NIH 3T3 cells were transfected with BAC DNA, and supernatant was collected a week later. Supernatants were passaged twice in NIH 3T3 cells before use. All tissue culture infection experiments were performed at an MOI of 1.

## Mice

BALB/cJ were purchased from The Jackson Laboratory. All mice were maintained under specific pathogen free conditions in the UC-Berkeley Animal Facilities. Mice used in experiments were between 3 and 8 weeks of age. All experimental procedures were conducted in accordance with the institution guidelines for care and use. Mice were infected with the indicated amount of virus. Liver, lung, and spleen homogenates were prepared at day five post infection, and viral titer was determined by qPCR specific for MCMV gB as described previously (*Smith et al., 2008*).

## Luciferase assay

Mouse fibroblasts were transfected with indicated constructs. Six separate transfections were averaged for each condition. At 24 hr post transfection passive lysis buffer (Promega #E1941) was used to lyse the cells. Lysates were transferred to an opaque assay plate (Corning, Corning NY) and D-Luciferin reagent was added to the plate. Luminescence was assessed over 10 s using an LMAX-II luminometer.

## RAE-1 staining

Mouse fibroblasts were harvested in 2 mM EDTA in PBS and stained with monoclonal rat anti pan-RAE-1or Rat IgG$_{2A}$ isotype control (R&D systems, Minneapolis MN) followed by APC-conjugated goat anti-rat IgG (Jackson ImmunoResearch, West Grove PA). All samples were co-stained with 7-

**Table 2.** BAC mutagenesis primers.

| Primer pair | Forward | Reverse |
|---|---|---|
| *Kanamycin** | CGATTTATTCAACAAAGCCACGTTGTGTCT | GCCAGTGTTACAACCAATTAACCAATTCTGA |
| Δ1-6 | gtgtcacgcgcacgtgttagcataggaatccagacgcgcgctcgcctgag | atttacatactcaggacaggtgtgggcggttccaggtgtacgtaagcaga |
| Δ6-12 | acacgcccaaaatcacgcaatcatatataaatggacaatgaagccaatct | gttctaagtaaaaggggatacgggcgggcgatacagatgtacgaacccaa |
| Δ12-18 | caacaaataaaaattgtacgctcattttatcgcgtctctgtcatgtgttc | gaggggttggtacggttcgagcgattttggtagtccgagacgtccgccgc |
| Δ18-22 | ttgaatacgattgttttttattggcagcactgagcacacgtccccccccacc | atccgctcgaggccatgctcaccaagaagaccgagtgtcccaacaacttc |
| Δ1-22 | gtgtcacgcgcacgtgttagcataggaatccagacgcgcgctcgcctgag | atccgctcgaggccatgctcaccaagaagaccgagtgtcccaacaacttc |
| Δ12 | caacaaataaaaattgtacgctcattttatcgcgtctctgtcatgtgttc | gttctaagtaaaaggggatacgggcgggcgatacagatgtacgaacccaa |
| Δ13 | tagaacaatatgtaaaccatctctcattcagctacatacagacaagggac | Tgataagaattatacttttaatgggggacacgttctagaacacgataaact |
| Δ14 | Agtataattcttatcaattataccagagtttggtattttttttaatctgag | Cgagtgtgaaatggggaaactggcgcgtcttttcattcgtgctccacagc |
| Δ15/16 | cactcgctatccttcgaccacactttcgagtgtctttttaccgtatcaagaag | Acaataaagatttcagacaaaaagtatggattgtgtgataatttattaaa |
| Δ17 | catactttttgtctgaaatctttattgtacgccatcgaaataaggggagc | gtctgctttctttgaaatcggacgaccgatcagaacgtccgccttcgaga |
| Δ18 | ttgaatacgattgttttttattggcagcactgagcacacgtccccccccacc | Gaggggttggtacggttcgagcgattttggtagtccgagacgtccgccgc |
| Δ19 | ccaagacgctcgtcttataacaccgactgacgtttactccgactcaggat | Tcgaggcgagtcttcggagctgtacgctagggcgatcgccatcaccctct |
| Δ20/21 | Cggcgacgacggcgatcacggcgagggtgaagagggtgatggcgatcgcc | Gctgtcatgtaaatggacggttattaaaagatgaggtcgtgtgacctctg |
| Δ22 | Gggtagcgcctcgatcgacgagcgtcggacaaagaaaccgggagaagaag | Atccgctcgaggccatgctcaccaagaagaccgagtgtcccaacaacttc |
| MCMV$^{m18stop}$ | gcagcggttccgccgtccccatcgcgacgatgggcgctccgaattcctaataa accgactcccgtccccaccaaggatgacgacgataagtaggg | ggagcgcccatcgtcgcgatttattaggaattccgcgctgctggcgatgagcgtggtg gggacgggagtcggtaaccaattaaccaattctgattag |

*Kanamycin specific oligos were added to the 3 the 3ggaattccgcgctgctggcgatgagcgtggtgggga.

AAD (BD Biosciences, San Jose CA) to exclude dead cells. Cells were analyzed by Flow cytometry using an LSR-Fortessa flow cytometer (BD Biosciences).

## Electrophoretic mobility shift assay (EMSA)

Nuclear lysates were prepared as described by Jianping Ye (Pennington Biomedical Research Center, Louisiana State University). Oligonucleotides for m18RE and Sp1 consensus sites can be found in *Table 3*. Oligonucleotides were labeled with $^{32}$P-γ-ATP using T4 kinase (NEB). Probes were purified on a G-50 column (G&E health care, Little Chalfont UK), and incorporated radioactivity was measured using a Beckman LS60001C scintillation counter. 4000cpm of labeled probe were added to nuclear lysates. Where indicated, competing unlabeled DNA probes were included in the reaction at a 1000:1 ratio. For super-shift assay 1ug of indicated Ab was added. Samples were run on a 5% native acrylamide gel. Gels were dried before being exposed in phosphofluor cassettes and analyzed using a Typhoon imager.

## Affinity purification and peptide sequencing by LC-MS/MS

Affinity-purification mass spectrometry (APMS) was used to identify candidate host-virus protein-protein interactions for the m18 protein. To this end, the annotated m18 orf was cloned into the pcDNA4TO expression vector encoding either an N-terminal or C-terminal 2X-StrepTag (m18-NS or m18-CS) for affinity purification and peptide sequencing by tandem liquid chromatography-mass spectrometry (LC-MS/MS) using methods identical to those previously reported (*Greninger et al., 2012*) Briefly, 10 µg of vector were transfected into 15 cm cultures of HEK293T cells using a Transit-LT1 reagent (Mirus Bio, Madison, WI) at a 3:1 vol to mg plasmid, and the cells were grown for 48 hr. Lysates were prepared in 0.1% NP40, 50 mM Tris HCl pH 8.0, 150 mM NaCl, 1 mM EDTA. The M18 protein was captured on StrepTactin Sepharose, and then eluted with 1X desthiobiotin (IBA Technology, Gottingen Germany) as reported. To identify captured proteins by proteomic analysis, the protein samples were reduced with DTT, alkylated with iodoacetamide, and digested in solution with sequencing grade porcine trypsin (Promega) following an identical protocol to that reported (*Greninger et al., 2012*). The resulting peptides were subjected to LC-MS/MS on an LTQ-FT mass spectrometer (Thermo Scientific) equipped with a Nano-Acquity ultraperformance liquid chromatography system (Waters) for reversed-phase chromatography with a C18 column (BEH130; 1.7 µm bead size, 100 µm by 100 µm), using identical acquisition parameters as reported (*Greninger et al., 2012*). MS data were searched using Protein Prospector software v. 5.10.17 (54) against the sequence of the m18 protein constructs and the NCBI Refseq human + virus database (downloaded Jan. 14, 2012) containing 131,459 sequences, concatenated with 131,459 additional randomized decoy sequences (*Chalkley et al., 2008*). A false discovery rate of <1% was obtained using protein score of 22, peptide score 15, protein expectation value 0.05 and a peptide expectation value of 0.001. Modifications allowed in the protein identification search were: fixed carbamidomethylation of Cys and the following variable modifications: oxidation of Met, start-Met cleavage, oxidation of the N-terminus, acetylation of the N-terminus, and pyroglutamate formation from Gln. HEK293T cells were chosen for these experiments to allow for identification of frequent background proteins and for specificity scoring, by comparison with a background dataset of unrelated picornavirus-host APMS experiments assayed in the same experimental system (*Greninger et al., 2012*, PMC4332878). Specificity scoring by Z-score was calculated using N = 3 m18 APMS experiments, which included one m18-NS and two biological replicate m18-CS experiments, and a background dataset of 598 unrelated picornavirus-host APMS experiments. Additional peptides for the m18

**Table 3.** EMSA Oligos.

| Primer pair | Forward |
| --- | --- |
| *m18RE* | ggctcgcaggtccacgcccttggcaccggag |
| *m18RE\** | ggctcgcaggtccaaacccttggcaccggag |
| *Sp Consensus* | attcgatcggggcggggcgagc |
| *Sp\* Consensus* | attcgatcggttcggggcgagc |

**Table 4.** Cloning and Mutagenesis primers.

| Primer pair | Forward | Reverse |
|---|---|---|
| *m18RE mutant Quickchange* | ggaggctcgcaggtccaaaccccttggcaccggag | ctccggtgccaaggggtttggacctgcgagcctcc |
| *m18* | atggctgacactgggc | tcaatcatcccaccagagag |
| *m19* | gatcgaattcATGAGTATCATCGCCACACCCATCC | gatcgcggccgcTCACCCTCGCCGTGATCG |
| *EBNA-1* | ATCGGAATTCGCCACCATGTCTGACGAGGGGCCAG | AATTCTCGAGCTCCTGCCCTTCCTCACC |

protein constructs were identified by allowing additional missed cleavages and Ser/Thr phosphorylation. Phosphorylation sites are reported with a site localization (SLIP) score, where SLIP $\geq6$ corresponds to >95% confident site assignment (PMC3134073).

## Western blotting

Nuclear and cytoplasmic lysates were separated as above for EMSA analysis. Protein amounts were quantified using BCA assay (Thermo). Cell lysates were run on a 4–12% SDS-PAGE gradient gel and transferred to Immobilon-fl PVDF membranes (Millipore). Membranes were blocked with 5% Milk in PBS with 0.05%, or 1% BSA Tween before being probed with the indicated antibodies. Where phosphor-epitopes were being assayed, 1% BSA was used in place of 5% milk. Membranes were probed with Li-COR secondary antibodies and imaged on an Odyssey Li-COR imager.

## CHiP

ChIP was performed essentially as previously described (*Elias and Gygi, 2007*) with the exception being the use of a Fisher Scientific Sonic Dismembrator Model 100 to shear chromatin. *Raet1e* Promoter DNA was quantified by qPCR using previously described primers (*Tokuyama et al., 2011*). All samples were analyzed in triplicate.

## Gene cloning

Primers to genes of interest were designed using cDNA sequences available in the Uniprot database. Primers can be found *Table 4*.

## Site directed mutagenesis

Site directed mutagenesis was carried out using the Quick-change site directed mutagenesis protocol (Stratagene, La Jolla CA). Primers used can be found below.

## Plaque assay

Plaque assays were performed as previously described (*Tokuyama et al., 2011*).

## Immunofluorescence assay

Mouse fibroblasts were plated onto glass slides before transfection with m18-RFP as described above. IFA was performed essentially as previously described using the following staining buffers (1%BSA in PBS for pHDAC1/3 and HDAC1/3, 1% Goat serum in PBS for AcH3 or H3) (*Bekerman et al., 2013*). Fluorescent signal intensity in the nucleus was quantified using FIJI software (*Karijolich et al., 2014*).

## Acknowledgements

The authors would like to thank Dr Hidde Ploegh for MCMVΔ1–22 mutant virus, Dr Martin Messerle for sharing MCMV BAC pSM3fr and the pDK46 plasmid, Dr Lewis Lanier for sharing MCA-205 carcinoma cell line, Dr Knipe for sharing the pICP0 plasmid, Dr Caroline Kulesza for sharing MCMV BAC pSM3fr in GS1783 e. coli, and Dr Leor Weinberger for sharing the pRSV-IE72 plasmid. The authors thank Andrew Birnberg for technical assistance. Mass spectrometry was provided through the UCSF

Mass Spectrometry Facility (AL Burlingame director), supported by (AL Burlingame, Director) supported by 8P41GM103481. JLD is supported by the Howard Hughes Medical Institute.

## Additional information

### Funding

| Funder | Grant reference number | Author |
| --- | --- | --- |
| National Institutes of Health | AI 100829 | Trever T Greene |
| National Institutes of Health | R01 AI113041 | David H Raulet |

The funders had no role in study design, data collection and interpretation, or the decision to submit the work for publication.

### Author contributions

TTG, MT, Conception and design, Acquisition of data, Analysis and interpretation of data, Drafting or revising the article; GMK, Acquisition of data, Analysis and interpretation of data, Drafting or revising the article; MK, JL, ALG, Acquisition of data, Analysis and interpretation of data; VRD, JLD, Conception and design, Contributed unpublished essential data or reagents; DHR, Conception and design, Drafting or revising the article; LC, Conception and design, Analysis and interpretation of data, Drafting and revising the article

### Ethics

Animal experimentation: This study was performed in accordance with the recommendations in the Guide for the Care and Use of Laboratory Animals of the National Institutes of Health. All animals were handled according to approved institutional animal care and use committee (IACUC) protocols of the University of California Berkeley. The specific animal use protocol (AUP#R292-0517BCR) was approved by the animal care and use committee (ACUC) of the University of California Berkeley. Every effort was made to minimize suffering.

## Additional files

### Supplementary files

• Supplementary file 1. Proteins identified by LC-MS/MS for M18-CS expressed in HEK293 cells. Reported here are the Protein Prospector data search results including number of unique peptides, the peptide count, and percent sequence coverage for a given protein identified, as well as best discriminant score and best expectation values. Protein Prospector data search parameters were reported in Materials and methods; a minimum of two unique peptides in at least one experiment were required for a protein to be reported. Known background proteins are reported in Supplemental Table 2 (after *Greninger et al., 2012*).

• Supplementary file 2. Frequent Background proteins identified in LC-MS/MS analysis of N- or C-terminally tagged M18 APMS samples. Reported here are the Protein Prospector data search results as in Supplemental Table 1. These background proteins appeared with ≥10% frequency among APMS experiments in our database of 2X-Strep-tagged protein expression in HEK293 cells for picornavirus-host AP-MS experiments (*Greninger et al., 2012*).

• Supplementary file 3. Z-scores for proteins identified by APMS. Proteins identified by APMS were scored for specificity by Z-score using a previously reported method (*Greninger et al., 2012*). N = 3 M18 APMS experiments included one N-term and two biological replicate C-term experiments. Background dataset included 598 unrelated picornavirus-host APMS experiments assayed in the same experimental system (*Greninger et al., 2012*).

• Supplementary file 4. LC/MS/MS identified peptides for m18. Provided here is a peptide report output from Protein Prospector, listing the LC-MS/MS identified peptides for m18 with N- or C-terminal 2X-Strep-Tag (sequences below). Provided are the observed mass to charge ratio (m/z) and charge (z) for each parent ion, with an error (in ppm) from the theoretical mass. Protein Prospector Peptide Score and Expectation Values are also provided. Variable modification sites are reported with protein numbering within the canonical M18 sequence using the '@' to indicate the position. A site localization probability (SLIP) score $\geq 6$ corresponds to >95% confident site assignment. Unambiguous SLIP scores are reported after the '=' symbol, and sites indicated with a vertical bar '|' show alternative matches that fit the spectrum.

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
