## [Decision Letter]

Thank you for submitting your article "A Herpesviral induction of RAE-1 NKG2D ligand expression occurs through release of HDAC mediated repression" for consideration by *eLife*. Your article has been reviewed by two peer reviewers, and the evaluation has been overseen by a Reviewing Editor and Tadatsugu Taniguchi as the Senior Editor.

The reviewers have discussed the reviews with one another and the Reviewing Editor has drafted this decision to help you prepare a revised submission.

Summary:

This is a well written and scholarly study describing that the RAE-1 family of NKG2D ligands expression pattern is repressed by phosphorylation and activation of HDAC3. The kinase has been identified as Casein Kinase 2. The study extends the known role of HDAC inhibiting virally encoded molecules. HDAC3 has been previously identified as a deacetylase target of DNA virus genes. Parts of this story have been previously described by the authors and other kinases have been known to modify the activity of deacetylases. The authors correctly refer to previous studies that document similar elements of the theme of regulation of enzymes that control expression of NK type ligands. However, they should cite and comment on a recent study {Mitotic Activation of a Novel Histone Deacetylase 3-Linker Histone H1.3 Protein Complex by Protein Kinase CK2. Patil H, et al. J Biol Chem. 2016 Feb 12;291(7):3158-72. doi: 10.1074/jbc.M115.643874. Epub 2015 Dec 9} which identified CK as a modifier of HDAC3 function and which affects chromatin structure similar to that proposed in this work.

There was some overall concern that the paper is too specialized for *eLife* because many of the elements (HDAC inhibitors in viruses, HDAC in regulation of NKG2D ligands, etc.) have been previously described, albeit in different contexts. However, the discovery of a specific inhibitor encoded by a DNA virus that affects NKG2D ligand expression through the specific mechanism described appears to be novel enough to sway the reviewers and editors positively at this point. That said, the reviewers had some major concerns.

Essential revisions:

Major concerns dealt with the in vivo relevance of the finding, the characterization of m18 protein and the mass spec data.

1) The authors should infect mice with the δ-m18 virus to examine if the absence of m18 affects viral virulence in an NKG2D-dependent manner. They should discuss such findings with respect to previous studies regarding NKG2D, NKG2D ligands, inhibitors, and MCMV virulence.

2) Insufficient characterization of m18 protein. The authors did not show the presence of the m18 protein in WB in spite of the fact that they have made numerous m18 constructs (m18-RFP, m18-GFP, m18-FLAG, m18-strep tag) and transfectants. Here are some examples: why were mouse fibroblasts, which are stably expressing m18 (Figure 5), analysed for the H3 and AcH3 protein expression and not for m18 in the same WB analysis? The same applies to the mouse carcinoma cells MCA-205 expressing m18. In addition, the authors used FLAG pull down from the lysates of cells expressing 3xFlag-m18 and tested for the presence of CK2β by WB, but again the m18 protein was not included in the analysis (Figure 6). The only example where m18 protein has been shown is the m18-RFP signal (Figure 5). There is no information about the m18 construct used for the m18 IF in Figure 6. Based on the nuclear localization and the term 'RFP-tagged', I assume that the construct was generated to produce a RFP/m18 fusion protein. Also, it would be important to address which of the fusion partners were positioned N terminally.

3) The mass spec analysis was not sufficiently elaborated. There was no information about the samples being analysed in the Results section. Also, as stated in the sentence: 'All reported m18 peptides were manually validated' it is unclear whether they were able to validate all predicted m18 peptides. When it comes to the native, virally derived, m18 protein so far only the m18 peptides have been confirmed (Holtappels et al., J Gen Virol, 2001; Kattenhorn et al., J. Virol, 2004).

---

## [Author Response]

[…]

*Essential revisions:*

Major concerns dealt with the in vivo relevance of the finding, the characterization of m18 protein and the mass spec data.

*1) The authors should infect mice with the δ-m18 virus to examine if the absence of m18 affects viral virulence in an NKG2D-dependent manner. They should discuss such findings with respect to previous studies regarding NKG2D, NKG2D ligands, inhibitors, and MCMV virulence.*

Extensive work has been done to characterize the role of m18 in MCMV virulence. Because the ∆m18 MCMV used in Figure 1 contains a kanamycin cassette and has the potential to produce confounding T cell epitopes in vivo, we generated and characterized an additional MCMV mutant to specifically eliminate the production of the m18 protein with minimal disruption of other viral genes. We accomplished this by inserting two stop codons into the m18 ORF near the 5’ end. This virus we referred to as MCMV^m18stop^. We then had to characterize MCMV^m18stop^ in vitrobefore the work could be

moved in vivo. As predicted, this virus failed to induce the expression of *Raet1* in vitro, and did not demonstrate any obvious defects in growth in vitrorecapitulating our original data using ∆m18. (This data is included as Figure 7—figure supplement 1).

Using this virus we answered two questions about m18 in MCMV biology in vivo.

Does m18 make MCMV more susceptible to NK cell control?

Is m18 required for growth of MCMV in vivo*?*

To answer these questions we infected MCMV susceptible BALB/c mice with this MCMV^m18stop^ virus and measured viral titers at five days post infection, as this is a time where NK surveillance significantly contributes MCMV replication in vivo. We performed this experiment three separate times with 5-7 mice per group to achieve significance and confidence in the results. Interestingly, we did not see any difference in MCMV levels in the lungs, liver or spleens of mice infected with MCMV^m18stop^ as compared to wild type MCMV. This suggests that m18 is not required for NK cell control or for MCMV replication in vivo at this time point.

We then sought to assess the growth of MCMV^m18stop^ in a system without surveillance by NK cells and other immune cells. To test this we infected RAG^-/-^RAG^-/-^ yC^-/-^ mice lacking NK cells, T cells, and B cells with MCMV^m18stop^ and measured viral titers at five days post-infection in the lungs, livers, and spleens. Again we did not see any difference in MCMV titers between wild type and mutant viruses in any of these organs.

Overall these data are in line with studies showing that viruses are competent for evasion of RAE-1 recognition by NKG2D (through m152) do not see a large increase in titers when NK cells are depleted, as this evasion is very efficient. Our model figure has been changed to clearly reflect this observation (Figure 8).

Beyond the analysis described above, we have preliminary data showing a large defect in MCMV^m18stop^ growth at five days in the salivary gland. These data do suggest that m18 is important for either trafficking of MCMV to the salivary glands or establishment within them. While this is very interesting it is highly likely that this phenotype is independent of RAE-1 induction by m18, as control of MCMV in the salivary glands is mostly dependent on CD4 & CD8 T cells. Thus, these data will be better served in a future study of m18’s function in the MCMV pathogenesis. In addition, it remains possible that m18 plays a role in additional aspects of MCMV pathogenesis that were not investigating here such as latency establishment, reactivation or transmission. This will be an interesting avenue to pursue in the future.

*2) Insufficient characterization of m18 protein. The authors did not show the presence of the m18 protein in WB in spite of the fact that they have made numerous m18 constructs (m18-RFP, m18-GFP, m18-FLAG, m18-strep tag) and transfectants. Here are some examples: why were mouse fibroblasts, which are stably expressing m18 (Figure 5), analysed for the H3 and AcH3 protein expression and not for m18 in the same WB analysis? The same applies to the mouse carcinoma cells MCA-205 expressing m18. In addition, the authors used FLAG pull down from the lysates of cells expressing 3xFlag-m18 and tested for the presence of CK2β by WB, but again the m18 protein was not included in the analysis (Figure 6). The only example where m18 protein has been shown is the m18-RFP signal (Figure 5). There is no information about the m18 construct used for the m18 IF in Figure 6. Based on the nuclear localization and the term 'RFP-tagged', I assume that the construct was generated to produce a RFP/m18 fusion protein. Also, it would be important to address which of the fusion partners were positioned N terminally.*

To address the reviewer’s concerns about our characterization of the m18 protein we have included a western blot demonstrating m18 protein expression in mouse fibroblasts and NIH-3T3 (Figure 2). Additionally, we have included an image of a fibroblast expressing a C-Terminal GFP m18 fusion protein illustrating that m18 localizes mostly to the nucleus (Figure 2). We have also added Figure 2—figure supplement 1 showing our characterization of the other constructs used in the manuscript. This includes western blot analysis of m18-3xFlag and m18-GFP, as well as flow cytometric analysis demonstrating that m18-3xFlag, m18-GFP, and m18-RFP all induce the expression of RAE-1. We have also included analysis correlating the levels m18-GFP to the levels of RAE-1 expression.

For Figure 5, we did not have an epitope tag in that construct, and there is no antibody for native m18 protein. We did however analyze the transduced populations for RAE-1 expression (Figure 3—figure supplement 2), and show that m18 transduced cells express high levels of RAE-1, similar to the effect of other m18 constructs used in the manuscript. Thus we are confident that these cells are expressing m18.

The lack of data illustrating m18 protein expression in our IP-WB experiment, was an oversight on our part. In the revised manuscript, we have included western blot analysis of m18Flag expression in the lysates of these fibroblasts Figure 6. We also include a new and extensive analysis of m18 peptides identified during our AP-MS purification demonstrating that the m18 protein was present in these samples. These data also indicate that m18 undergoes extensive post-translational modification as 45% of peptides identified contained some type of modification.

Additionally, we have revised the manuscript to clearly indicate which terminus each epitope tag or fluorescent protein was fused to for each individual experiment.

*3) The mass spec analysis was not sufficiently elaborated. There was no information about the samples being analysed in the Results section. Also, as stated in the sentence: 'All reported m18 peptides were manually validated' it is unclear whether they were able to validate all predicted m18 peptides. When it comes to the native, virally derived, m18 protein so far only the m18 peptides have been confirmed (Holtappels et al., J Gen Virol, 2001; Kattenhorn et al., J. Virol, 2004).*

To address the reviewer’s concerns we have extensively elaborated upon our mass spec analysis in both the Results and Methods. The Results section has been improved to indicate that AP-MS was performed using N- or C-terminally Strep-tagged m18 was transiently expressed in HEK293T cells to allow for direct comparison with similar virus-host AP-MS samples. This allowed for high frequency background proteins to be identified. We also indicate that this experiment was repeated multiple times.

We have included four new supplementary files tabulating our MS results (Figure 6—figure supplements 1-4). This includes specific interacting proteins identified in the APMS ([Supplementary-material SD1-data]), high frequency background proteins identified in the APMS ([Supplementary-material SD2-data]), Z scores for ranking the most specific interacting proteins identified in the APMS ([Supplementary-material SD3-data]), and a list of all identified peptides (including post translational modifications) sequences derived from the annotated m18 ORF ([Supplementary-material SD4-data]). The phrase “manually validated” was not meant to be confusing, and is a term used often in proteomic groups to indicate that peptide spectra were inspected for the quality of the matched peptide sequen, but more importantly here we have provided detailed explanation of the scoring used for peptide sequence identifications by LC-MS/MS.

In our Methods we have changed our wording to indicate that the *reported*peptides referenced in the original manuscript were identified by searching a database that contained the predicted annotated m18 ORF. For the interest of the reviewer we did identify an m18-derived peptide that overlaps the immunogenic peptide identified in Holtapples et al. 2001 (QKAASAASSSSSASSSGPSR), but we were unable to confirm how our peptides related to those identified in Kattenhorn et al. 2004, as the identity of the peptides identified for m18 in this study were not publicly available.